# Perfusable Tissue Bioprinted into a 3D-Printed Tailored Bioreactor System

**DOI:** 10.3390/bioengineering11010068

**Published:** 2024-01-09

**Authors:** Marius Gensler, Christoph Malkmus, Philipp Ockermann, Marc Möllmann, Lukas Hahn, Sahar Salehi, Robert Luxenhofer, Aldo R. Boccaccini, Jan Hansmann

**Affiliations:** 1Department Tissue Engineering and Regenerative Medicine, University Hospital Wuerzburg, 97070 Wuerzburg, Germany; 2Institute of Medical Engineering Schweinfurt, Technical University of Applied Sciences Wuerzburg-Schweinfurt, 97421 Schweinfurt, Germanyjan.hansmann@thws.de (J.H.); 3Translational Center for Regenerative Therapies (TLC-RT), Fraunhofer Institute for Silicate Research (ISC), 97082 Würzburg, Germany; 4Institute for Functional Materials and Biofabrication, Department of Chemistry and Pharmacy, Julius-Maximilians-University Wuerzburg, 97070 Würzburg, Germany; 5Department of Biomaterials, Faculty of Engineering Science, University of Bayreuth, 95447 Bayreuth, Germany; 6Institute of Biomaterials, University of Erlangen-Nürnberg, 91058 Erlangen, Germany

**Keywords:** biofabrication, 3D-printing, perfusable bioreactor, cell culture simulation, adipose tissue, bioprinting

## Abstract

Bioprinting provides a powerful tool for regenerative medicine, as it allows tissue construction with a patient’s specific geometry. However, tissue culture and maturation, commonly supported by dynamic bioreactors, are needed. We designed a workflow that creates an implant-specific bioreactor system, which is easily producible and customizable and supports cell cultivation and tissue maturation. First, a bioreactor was designed and different tissue geometries were simulated regarding shear stress and nutrient distribution to match cell culture requirements. These tissues were then directly bioprinted into the 3D-printed bioreactor. To prove the ability of cell maintenance, C2C12 cells in two bioinks were printed into the system and successfully cultured for two weeks. Next, human mesenchymal stem cells (hMSCs) were successfully differentiated toward an adipocyte lineage. As the last step of the presented strategy, we developed a prototype of an automated mobile docking station for the bioreactor. Overall, we present an open-source bioreactor system that is adaptable to a wound-specific geometry and allows cell culture and differentiation. This interdisciplinary roadmap is intended to close the gap between the lab and clinic and to integrate novel 3D-printing technologies for regenerative medicine.

## 1. Introduction

Like Tissue Engineering (TE) and Regenerative Medicine (RM), biofabrication focuses on the production of functional tissues and organs. Therefore, cells and scaffolds as building blocks are assembled by additive manufacturing techniques. Additive manufacturing has evolved as an efficient method to produce plastic and metal parts. By facilitating a defined deposition of cells and materials for the generation of biological tissues, biofabrication forms a synergetic intersection between additive manufacturing and life sciences [1,2].

Techniques for biofabrication are closely related to the most common three-dimensional (3D) printing techniques, i.e., Fused Deposition Modeling (FDM), Stereolithography (SLA), and Selective Laser Sintering (SLS). For FDM, solid filaments are fed into a heated printhead and squeezed through a nozzle after a heat-induced increase in the viscosity of the filament. To create a part, the melted filament is deposited onto a plate that is position-controlled in the x-, y-, and z-directions. In contrast, SLA uses photosensitive resins, which are crosslinked by a laser, whereas SLS locally melts a powder material using a laser. The melted material fuses and creates a part layer by layer. For biofabrication, the most common technique is material extrusion, which generates 3D constructs by the deposition of hydrogels or bioinks and is thus similar to FDM printing [2,3,4].

The properties of a bioink have a strong impact on the quality of a biofabricated construct. Hence, several requirements need to be considered [5]. First, the bioink must be printable: the material should ensure a suitable viscosity to support the geometrical integrity after printing, whereas high shear forces need to be prevented during the printing process to support cell survival. Additionally, the ink should provide a suitable microenvironment to allow controlled maturation of the printed tissue. This includes cell attachment, proliferation, differentiation, cell-driven re-organization of the scaffold, as well as a sufficient nutrient supply. Complementary, reasonable economic costs and a constant composition and quality between different batches qualify a material for biofabrication applications. Nevertheless, the prevention of shrinkage of the 3D-printed construct due to scaffold degeneration and cell migration is a common challenge in biofabrication [6]. One possibility to overcome this limitation of most current bioinks is improved hydrogel stability by crosslinking. Different mechanisms of crosslinking are available, such as chemical crosslinking using, e.g., calcium ions, and physical crosslinking by UV light or temperature [7]. Crosslinkable organic polymers like nanocellulose, alginate, gelatin, and collagen have been used for bioink generation [8]. To achieve suitable printability and stability as well as controlled tissue maturation, a variety of additives such as thermosensitive materials have been successfully supplemented to bioinks [9,10].

Since TE and RM strive for patient-specific tissues and geometries, bioprinting is a tool to fit a tissue construct to a patient’s defect individually. This has already been achieved for bone and skull implants, as well as soft tissues [11]. However, bioprinting is only the first step in the fabrication of biological implants. A subsequent culture under physiological conditions allows tissue maturation, including cell differentiation and matrix remodeling, thereby gaining tissue-specific functionality [12]. These culture conditions are provided by bioreactor systems.

Bioreactors are containers for tissue maintenance under controlled conditions, protected from contaminations and harmful external influences. Therefore, the applied bioreactor material needs to be biocompatible and chemically inert to not interact with the scaffold, cells, or medium. Furthermore, the material has to comply with cell culture requirements, such as temperature and humidity resistance, washability, and sterilizability. Moreover, bioreactor systems ensure a proper nutrient supply and removal of waste products by dynamically perfusing the tissue construct with fresh medium. Mechanical or electrical stimulation facilitates a more in vivo-like environment supporting a tissue-specific maturation [13,14].

Typically, bioreactor systems for tissue engineering have been designed to be used for multiple tissues and geometries, thereby reducing manufacturing costs and enhancing versatility [15,16,17,18,19]. However, common subtractive manufacturing methods applied for the generation of bioreactor components do not support the high degree of freedom that is required to fit an implant to a patient’s defect. A solution would be to apply additive manufacturing for both the generation of the biofabricated construct and the production of a tailored implant-specific bioreactor.

This work aims to design a workflow for the creation of an implant-specific dynamic bioreactor system that is (i) easily available and producible, (ii) customizable to geometric requirements, (iii) in compliance with standard cell culture requirements, and (iv) able to mature tissues in a dynamic system and thereby allowing seamless integration of tailored bioreactor technology into the biofabrication process. Thereby, guidance for the implementation of current 3D-printing technologies in clinical applications is envisioned.

## 2. Methods

### 2.1. Design of Tailored Components

Parts were designed using computer-aided design (CAD) with Solidworks^®^ Premium 2017 (Dassault Systèmes, Paris, France) and transferred into an STL file.

### 2.2. 3D Printing and Post-Processing

FDM printing was performed by transferring the STL file into a machine-specific Gcode using ideaMaker software (version 3.4.2-4.2.0; Raise3D Technologies Inc., Irvine, CA, USA). Printing parameters were set to 0.4 mm nozzle diameter, 200 μm layer height, 220 °C printing temperature, 60 °C printbed temperature, 50 mm/s printing speed, and 33% honeycomb infill ratio and type. Parts were printed with Green-TEC Pro Filament–Nature (GTP175X800NAT, FD3D GmbH, Lauterach, Austria) using a Raise3D Pro 2 printer (Raise3D Technologies Inc., USA).

For SLA, STL files were transferred to machine-specific Gcode using Preform (Formlabs Inc., Somerville, MA, USA). For printing parameters, the default settings of the materials were applied. Layer height was set to 50 μm. Parts were then printed with the Form 2 printer (Formlabs Inc., USA). After printing, the parts were washed with isopropanol (6752.5, Carl Roth GmbH, Karlsruhe, Germany) according to the company’s protocol using the Form Wash device (Formlabs Inc., USA) and subsequently cured with UV light in the Form Cure device (Formlabs Inc., USA) according to the default protocol and settings.

### 2.3. Sterilization

Sterilization was performed by either autoclaving at 121 °C for 15 min using a DX-45 autoclaving device (Systec GmbH, Hörstel, Germany) with a total process time of 1.5 h, or vaporized hydrogen peroxide plasma treatment (H_2_O_2_) with a Pico P100 (Diener electronic GmbH & Co. KG, Ebhausen, Germany) as referred to in [20]. Plasma-sterilized parts were not used for direct cell contact experiments within the first two weeks after sterilization.

### 2.4. Biocompatibility Test

Examination of the biocompatibility was adapted from DIN EN ISO 10993-5 [21]. CellTiter-Glo^®^ Luminescent Cell Viability Assay solution (G7570, Promega, Walldorf, Germany) was applied and luminescence was quantified with a Tecan plate reader (Tecan Trading AG, Männedorf, Switzerland). For analysis, the positive control was defined as 100%.

### 2.5. Cell Culture

The C2C12 cell line was cultured and expanded in DMEM (61965026, Thermo Fisher Scientific, Waltham, MA, USA) with 10% FCS (S 0615, Merck, GER, Darmstadt, Germany) by seeding 1000 cells per cm^2^ into cell culture T-flasks (90151, TPP, Trasadingen, Switzerland). The culture medium was exchanged every second to third day. At a confluency of 70%, cells were passaged.

Human mesenchymal stem cells (hMSCs) were cultured in DMEM/F12 (31331028, Thermo Fisher Scientific, Waltham, MA, USA) with 10% FCS (FBS.EUA.0500, Bio&Sell, Nürnberg, Germany) and 1% penicillin/streptomycin (P4333, Sigma, St. Louis, MO, USA). For adipose differentiation, the medium was changed to DMEM High Glucose (61965026, Thermo Fisher Scientific, USA) supplemented with 10% FCS, 1 µM dexamethasone (D4902, Sigma, USA), 1 µg/mL insulin (I9278, Sigma, USA), 100 µM indomethacin (I8280, Sigma, USA), 500 µM IBMX (A0695, Applichem, Darmstadt, Germany), 1% D-glucose (G8769, Sigma, USA), and 0.1% lipid mix (L0288, Sigma, USA).

### 2.6. Gcodes for Bioprinting

Gcodes for bioprinting were manually generated with Repetier-Host software (version 2.2.2; Hot-World GmbH & Co. KG, Willich, Germany). For both bioinks (Alginate-POx and Cellink^®^ Bioink), the printing parameters were set as follows: first layer height: 400 μm, following layer heights: 600 μm, strand distance: 1 mm, and the printing speed: 600 mm/min. Gcodes were manually adapted to printer-specific commands.

### 2.7. Bioink Preparation

The thermogelling AB diblock copolymer (Me-PMeOx100-b-PnPrOzi100-EIP), further referred to as ‘POx’, comprising the hydrophilic poly(2-methyl-2-oxazoline) (PMeOx) block and thermoresponsive poly(2-n-propyl-2-oxazine) (PnPrOzi) block was synthesized as described by Lorson et al. [22]. Derived from Hu et al. [23], Alginate–POx bioink was generated by mixing 25 weight % POx-polymer and 1 weight % alginate in PBS^-^ (diluted with Millipore water by a ratio of 1:3) overnight at 4 °C.

To prevent poly(2-oxazoline) from crosslinking, it was kept on ice during cell preparation. The required number of cells was resuspended in 100 µL PBS^-^, mixed with the bioink and subsequently stored at 37 °C for 30 min for pre-crosslinking of the poly(2-oxazoline). For C2C12 cells, the assigned concentration was 1 × 10^7^ cells/mL, and for hMSCs, it was 4 × 10^6^ cells/mL.

Commercial nanocellulose-based Cellink^®^ Bioink (IKC200000303, Cellink AB, Göteborg, Sweden) was mixed with the required number of cells and resuspended in 100 µL PBS^-^. The mixed bioink was then directly used for bioprinting.

### 2.8. Bioprinting

Inkredible+^TM^ bioprinter (Cellink AB, Sweden) was applied for bioprinting. The pressure was adjusted according to the used bioink (47–70 kPa for Alginate–POx and 35–60 kPa for Cellink^®^ Bioink). A 3D-printed adapter plate was designed to hold the 3D-printed tissue container. Due to the heterogeneous behavior of the inks and temperature fluctuations, the printing pressure had to be readjusted during the process individually.

Once the printing process was completed, the biofabricated construct was covered with 1–2 mL crosslinking agent (CL10100, Cellink AB, Sweden) for 30 min. The tissue container was then transferred to the bioreactor system for dynamic culture of the biofabricated construct.

### 2.9. CFD Simulation

Computational Fluid Dynamics (CFD) simulations were performed using the finite element method software COMSOL Multiphysics (version 5.3; COMSOL Multiphysics GmbH, Göttingen, Germany). The mesh size was set to extremely fine. The material of the tissue and medium part was defined as water. For characterizing the medium flow velocity as well as the shear stress, a built-in Reynolds-averaged Navier–Stokes k-turbulence flow model was used in a steady-state study. The medium mass inflow was set to 1.5 g/min.

Glucose concentration within the tissue was calculated by using the built-in transport of diluted specimen (TDS) model in a steady-state study. The initial concentration of the tissue part was set to 0 mol/m^3^ and the medium to 25 mol/m^3^ (equals 4.5 mg/L glucose in DMEM medium). The diffusion coefficient of glucose in water was taken from Stein et al. [24] and set to 6 × 10^−10^ m^2^/s. The elimination rate of the glucose was adapted from Ahn et al. [25] and calculated as −1.157 + 10^−4^ mol/(m^3^·s) (glucose consumption of 3 × 10^6^ cells/mL).

### 2.10. Production of Silicone-Based Bioreactor Parts

Bioreactor parts were made from poly(dimethylsiloxane) (PDMS) as previously published [26]. Briefly, the required structure was designed by CAD and 3D SLA-printed using the resin Model V2 (Formlabs, USA). A negative mold was created by using the two-component silicone Dublisil^®^ 15 (Dreve Dentamid GmbH, Unna, Germany). After plasma activation of the mold surface using a plasma cleaner device (Pico LF PC 115656, Diener electronic GmbH & Co. KG, Ebhausen, Germany), the two-component silicone PDMS (SYLGARD^TM^ 184, Dow Europe GmbH, Wiesbaden, Germany) was mixed 10:1 (pre-polymer and cross linker) and cast into the silicone mold. After overnight storage at 37 °C in an oven (HERATHERM, Thermo Fisher Scientific, USA), the cast PDMS part was removed from the Dublisil mold and autoclaved once before final use to ensure proper crosslinking.

### 2.11. Dynamic Culture

For dynamic tissue culture, a custom-made incubator was used [27]. Humidity control was not implemented in this system. After the bioreactor was assembled and filled with medium in a cell culture hood, it was placed into the incubator and mounted to a built-in peristaltic pump. The pump speed was then set to 5 rpm, equal to 1.5 mL/min. The culture media were exchanged weekly.

### 2.12. Assessment of Cell Viability

For fluorescent live/dead characterization of the bioprinted constructs, the LIVE/DEAD^TM^ Viability/Cytotoxicity Kit (L3224, Life Technologies Corp., Carlsbad, CA, USA) was used according to the manufacturer’s protocol. The constructs were then visualized using a fluorescence microscope equipped with a 470 nm filter for calcein (viable cells) and a 545 nm filter for ethidium homodimer (dead cells). Four representative areas of the same size were selected from the images obtained, recognizable dead and viable cells were counted, and the percentage of cell viability was calculated.

Qualitative viability was analyzed with a 3-(4,5-dimethylthiazol-2-yl)-2,5-diphenyltetrazolium bromide (MTT) (M2128, Sigma-Aldrich, Germany) assay. Constructs were submerged in MTT solution (1 mg/mL in cell culture medium) for 3 h at 37 °C. MTT solution was discarded, and the constructs were washed twice with PBS^+^.

### 2.13. Histological Analysis

Constructs were fixed with Roti-Histofix^®^ 4% (P087.3, Carl Roth, Karlsruhe, Germany), supplemented with 0.3 M CaCl_2_ to prevent de-crosslinking, on a rocking shaker at RT for 3 h (1 h per mm of thickness). Subsequently, samples were processed for paraffin embedding followed by hematoxylin and eosin staining of microsections.

### 2.14. Statistical Analysis

Calculations and statistical analysis were conducted using GraphPad Prism 9 (version 9.2.0; GraphPad Software, La Jolla, CA, USA) and Microsoft Excel (Microsoft, Redmond, WA, USA). The data and values are visualized as mean ± standard deviation.

## 3. Results

Based on a previously published guideline [20] for the development of a bioreactor that can be produced by additive manufacturing, a workflow concept for the sequential 3D printing of a bioreactor and biofabricate was developed (Figure 1). The individual wound geometry needs to be identified, i.e., by MRI, to then derive the affected part by CAD. In the next step, a defect-specific container and the referring tissue are printed. Following the printing steps, the tissue is matured in the tailored bioreactor system until necessary tissue properties and functions are achieved. The functional tissue can then be implanted into the patient.

The bioreactor geometry was aligned to a rapid prototyping approach and was limited to the defect-specific container harboring the biofabricate. The tailored container is enclosed in standard PDMS housing to seal and protect it from contamination. The PDMS housing serves as a platform and by adapting the FDM-printed container, different biofabricate geometries can be cultured. This concept facilitates a minimum printing time for the FDM process (Figure 2A). The PDMS housing itself had two compartments: a tissue chamber and a medium reservoir. An air filter allows pressure equalization and gas exchange. To improve the tightness, pressure was applied to the PDMS housing using a 3D-printed rack (Figure 2B,C). The medium flows vertically from the bottom to the top of the tissue chamber and thus through the contained tissue (Figure 2D). The tissue container is lignin based. The applied materials were tested for biocompatibility (Appendix A).

Furthermore, the bioreactor design had the following characteristic features (Figure 2E): An undercut geometry at the reservoir lid prevents capillary effects and adhesion, therefore enabling droplet formation (see Appendix A). The lid and housing have an interlocking closure to seal them tightly together. The medium reservoir has a level indicator to show medium level; each line indicates 2 mL. Additionally, the reservoir has a curved bottom to create a sinkhole that counteracts dead slipstream areas and allows the reservoir to be emptied completely.

As a proof of concept, different tissue geometries and their respective containers were designed (see Figure 3). The according Gcode was refined stepwise using Cellink^®^ Start hydrogel as it provides adequate bioink properties while being comparatively economical. For bioprinting, a nozzle with a diameter of 410 µm was used at a printing speed of 600 mm/min. The first layer height was set to 400 µm and to 600 µm for the following layers. The strand distance was 1 mm. Different geometries were directly printed into the respective containers (Figure 3, Appendix A).

To enable a proper nutrient supply in the biofabricate, the integration of tubular structures is required. Therefore, CFD simulations were applied (Figure 4), allowing the assessment of critical factors, such as shear stress and nutrient concentration, for a given tube configuration. As expected, shear stress was reduced with an increased number of channels in the tissue. To gain information about nutrient distribution, the glucose concentration was calculated. The diffusion coefficient and glucose elimination rate were taken from the literature [24,25]. The geometries Square4 and Flower5 showed adequate results for both shear stress in the channels (<10 mPa) and nutrient distribution throughout the whole tissue (>15 mol/m^3^). Due to the relatively high inner diameter, the Column also showed a comparably low shear stress. The glucose distribution, however, was reduced at the outer rim of the tissue. As fresh medium arrives from only one side, it is important to determine whether nutrient supply will occur over the full thickness of the tissue. Therefore, cut views of the glucose concentration were prepared and analyzed. Again, Square4 and Flower5 showed an adequate glucose distribution over the whole tissue, ranging from 15 to 24 mol/m^3^ (Figure 4), compared to Square1 and Flower1.

Overall, Flower5 showed both an adequate glucose concentration in the whole tissue and low shear stress in the channels, while being comparatively complex in geometry. Therefore, this geometry was used in further proof-of-concept experiments.

To assess basic requirements such as biocompatibility, cell survival, and handling and robustness of the system, as well as usability/applicability thereof, C2C12 cells were printed with a concentration of 1 × 10^7^ cells per ml in two different bioinks, Cellink^®^ Bioink (35–60 kPa) and POx–Alginate (47–70 kPa) (Figure 5A,B), and cultured for 14 days. The general setup for dynamic perfusion is shown in Figure 5C (see also Appendix A). Figure 5F,G show a qualitative viability test via MTT, which proves the cells are alive after 14 days of culture, further proving a sufficient nutrient distribution throughout the whole tissue. Figure 5D,E show that macroscopically, no shrinkage of the tissues occurred, and the pores were still aligned with the surrounding container. Overall, characteristics such as the flower shape and the channels were still macroscopically visible.

Figure 6A,C show a combinatory microscopic overview of the live–dead analysis and the HE staining for the Cellink^®^ Bioink; cells were visibly aligned to the deposition pattern of the bioprinting process in the fluorescence pictures (Figure 6A). However, inside the deposition pattern, the cells were evenly distributed and viable. The same holds true for the areas around the central and peripheral channels, and the tissue areas (Figure 6E).

For the Alginate–POx bioink, viable cells could be found throughout the whole tissue with an accumulation at the outer rim and around the central channel (Figure 6B,D). Higher magnifications of the central and peripheral channel, as well as the tissue area, revealed few dead cells with most cells being positive for calcein (Figure 6F; see also Appendix A for quantitative analysis).

Overall, these results show the ability of a 3D-printed bioreactor system to maintain cells over a period of two weeks.

The culture of the C2C12 biofabricates, based on both bioinks, proved the basic principle of tissue maintenance in dynamic culture within the 3D-printed bioreactor. To investigate tissue maturation, hMSCs were differentiated into adipocytes in the Cellink^®^ Bioink. Therefore, hMSCs were 3D printed with a density of 4 × 10^6^ cells/mL and were dynamically cultured for 3 weeks. Figure 7A,B show a combinatory microscopic overview of the live–dead analysis and the HE staining. As shown for the C2C12 biofabricate based on the Cellink^®^ Bioink, viable cells are aligned to the deposition pattern of the bioprinting process. Crucial structures, such as the flower shape and central and peripheral channels, are well recognizable. The structures are shown with a higher magnification in Figure 7C–H. The viability analysis imaged by fluorescent microscopy reveals a weak signal of dead cells, whereas the majority of hMSCs were positive for calcein (see also Appendix A for quantitative analysis). The HE staining of these structures shows differentiating hMSCs. Differentiating adipocytes store lipids in an increasing number of cytoplasmic vacuoles surrounding the cell nuclei. This multilocular morphology can be recognized for all cells embedded in the bioink. This result shows the ability of the system to maintain and differentiate hMSCs toward the adipogenic lineage under dynamic culture conditions.

The workflow for the generation of defect-specific implants (Figure 1) necessitates that the biofabrication and tissue maturation are performed in an expert laboratory. To support the transfer of the matured tissue implant to the patient in a clinical environment, a prototype of an automated mobile docking station for the tailored bioreactor system was developed as a proof of concept. This mobile system had to fulfill the following requirements: 1. sterility, 2. maintenance of appropriate dynamic culture conditions, 3. (automatic) media exchange, 4. sufficient independent power supply for transport duration, and 5. control panel for handling by non-laboratory staff. The circuit and flow diagram are added to the supplementary data (Appendix A). Figure 8 shows the docking station for the bioreactor system, based on the open-source microcontroller Arduino UNO. The device is capable of exchanging the culture medium automatically on demand, which is achieved by pinch valves and sterile ports for new and old media (Figure 8A). Due to its integrated power supply, it is able to work independently from other power sources for 3 h. Together with the integrated pump with adjustable speed and its comparably small size, this makes the device highly mobile.

## 4. Discussion

This study presents a workflow for the application of current 3D-printing technologies for the generation of a versatile bioreactor that can be adapted to defect-specific geometries. Based on this workflow, a roadmap for the implementation of biofabrication into a clinical environment was derived as illustrated in Figure 9.

The clinical application of a novel technology, however, requires certain prerequisites, which need to be met prior to the clinical test phases and eventually the implantation into human patients. These challenges are of biological and technical relevance. Biologically, typical cell culture components do not fulfill the requirements for clinical application and thus would need to be adapted, e.g., the applied bioinks in this study are not developed for clinical use. Also, reagents used for the culture and maturation of the tissue cannot contain materials of animal origin, such as FCS. Additionally, the materials used for the bioreactor have to be biocompatible so as not to harm the cells (Appendix A) [20]. The transplantation of allogeneic organs and tissues from a donor bears the risk of rejection [28]. The isolation of patient-specific cells for the biofabrication of a graft will prevent rejections by the immune system [28,29]. Notably, in the case of adipose tissue, allogeneic transplantation of cells is enabled by the immunosuppressive properties [30]. In vitro pre-differentiated adipose tissue grafts were transplanted without causing rejection [31]. In the study, pre-differentiated adipocytes were subcutaneously injected. Using 3D-printing technologies, biofabrication enables a flexible production environment for defect-specific grafts with the required geometry.

This study focused on the technical base for the biofabrication roadmap to ensure a methodologically sound technology. Therefore, technical requirements (aims i–iv) were identified and measures were developed. To ensure that the bioreactor system is easily available and producible, it can be recreated using PDMS casting and FDM printing, both being commonly available. These basic manufacturing techniques enable its production in competent laboratories worldwide (aim i). All necessary design and construction files are available for public use (Appendix A). For the generation of a bioreactor system, the applied materials need to be able to withstand conditions necessary for cell culture, such as sterilization temperatures or high humidity. PDMS offers a cost-efficient, easily available, but nevertheless chemically inert and biocompatible solution [32]. Also, the PDMS transparency simplifies cell culture by providing visual insights into an otherwise closed system. Color changes in the culture medium as an indicator of nutrient deprivation or contamination may therefore be detected promptly.

The flexibility of the system to varying tissue geometries is achieved by the customization of the FDM-printed tissue container (aim ii). This principle allows the adaption of a single component to the defect-specific geometry, in which the tissue is bio-printed (see Figure 3). This further promotes the time efficiency for geometry adaptions and the cost effectiveness of the tailored bioreactor system. Furthermore, FDM printing can be integrated into commonly applied bio-printers, which enables the opportunity of simultaneous printing within a single device (see i.e., [33]). To ensure a sufficient nutrient supply in the defect-specific bio-printed tissue, different patterns of channels were created and analyzed via in silico studies of medium flow, shear stress, and nutrient concentrations. The simulation provides information on how to adapt the geometry to the patient-specific defect and the design of the channel architecture (Figure 4) to optimize the nutrient supply. With this, the system is scalable to bigger tissue sizes. Notably, bioprinting inside a 3D-printed defect-specific container could be shown for the first time.

As proof of concept for the ability of the system to maintain a bio-printed tissue, murine C2C12 cells in POx–Alginate and Cellink^®^ Bioink were cultured for 14 days. The high cell viability demonstrated a sufficient nutrient distribution in the tissue without harmful impacts of the materials or contaminations. There was no macroscopical shrinkage of the tissues and cell aggregation around the pores (aim iii). Furthermore, we were able to differentiate hMSCs into the adipocyte lineage, whose multilocular morphology could be recognized in the histological analysis (Figure 7). This showed the ability of the system to maintain and differentiate hMSCs adipogenically. Overall, the dynamic culture within the tailored bioreactor allowed the maturation of the tissue, specifically the differentiation of cells (aim iv). Typically, matured adipocytes are characterized by their unilocularity, a single lipid vacuole, and a distinct marker profile [34]. The in vitro maturation might be achieved in further experiments by the prolongation of the culture period up to three months [35]. Nevertheless, the system’s capability to differentiate cells and mature tissues in general could be proven.

For this study, we used a commercially available bioink and a blend composed of alginate and POx, both showing the required features for printing [22,23]. The POx–Alginate bioink was included in this study to allow a head-to-head comparison with the commercial bioink. Previously, we have shown that the combination of alginate and the POx-based diblock copolymer hydrogels significantly improves the 3D printability of alginate and allows for long-term cell culture [23]. Now, we wanted to compare it directly with a commercial bioink to gauge the potential of our experimental bioink. While alginate was used for scaffold stability after printing, POx was dedicated to not only ensure a sufficient temperature-dependent viscosity [23] during the printing process but also serve as a sacrificing structure after printing. A blended bioink has the advantage of being more easily adaptable to a specific use case (e.g., temperature, viscosity, or pore size). Both bioinks allowed the generation of biofabricates containing channel structures in the scaffold for nutrient supply. This suggests our approach is also adaptable for other commercial or blended bioinks, further extending the use cases.

A compatible docking station facilitates the parallelization of tissue maturation in a non-expert lab facilitated by low handling requirements and a high degree of automation. The developed docking station has the advantage of eliminating the need to disassemble the system for medium changes, lowering the risk of contamination and reducing the amount of time without perfusion. Furthermore, the bioreactor system can be run in a sterile environment (i.e., a clean bench) for three hours independently. Experiments in the combined bioreactor and docking station system are thus more standardized than experiments in the bioreactor system alone.

Technically, the devised docking station represents a developmental stage, which is not suitable for a clinical environment as it is not designed to fulfill the technical requirements regarding safety standards yet and rather serves as a proof of concept in this study [36,37].

In summary, we provided a roadmap (Figure 9) for the future applications of biofabrication in a defect-specific geometry using technologies that are commonly available (aim i). The system was designed to allow maximum geometric flexibility (aim ii), tissue maintenance (aim iii), and maturation represented by cell differentiation (aim iv).

## 5. Conclusions

Our proposed roadmap provides a future strategy for the implementation of biofabricated, defect-specific grafts until the clinical application. For its achievement, current technologies of 3D printing and bioprinting represent the basic methodologies. These technologies allow a maximal variance regarding the geometric requirements of a defect and relatively low-technical equipment. Furthermore, the docking station enables automated handling by non-experts. It represents an important step from bench to bed, thereby allowing the application of translational medicinal technologies for personalized medicine.

## Figures and Tables

**Figure 1 bioengineering-11-00068-f001:**
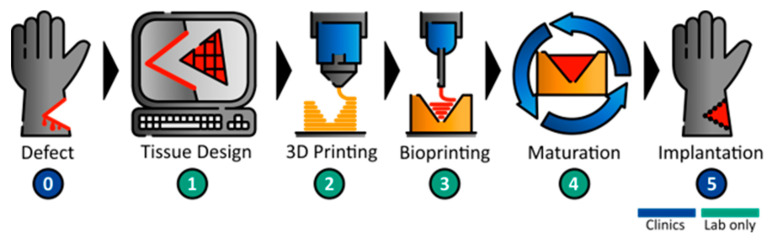
**Workflow for 3D printing of customized bioreactors and defect-specific bioprinted tissue constructs.** After identification of defect geometry (**0**), the affected part is redesigned by CAD (**1**). According to the tissue geometry, a tailored tissue container is designed and 3D printed (**2**). Afterward, the tissue is bioprinted into the defect-specific tissue container (**3**), followed by tissue maturation in a bioreactor system (**4**). Finally, the functional tissue can be implanted into the patient (**5**).

**Figure 2 bioengineering-11-00068-f002:**
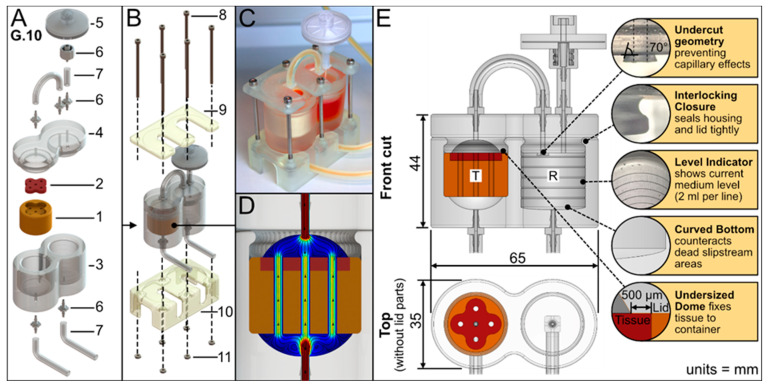
**3D-printed bioreactor setup.** (**A**) Exploded view of the bioreactor parts: A preprinted tissue container (**1**) holding the bioprinted tissue (**2**) is inserted into the bioreactor housing (**3**) made from PDMS. The housing mainly forms two compartments: the tissue chamber, and the medium reservoir. A lid (**4**) made from PDMS seals the bioreactor. Silicon tubes (**7**) as well as a sterile air filter (**5**) are connected to the bioreactor via LUER connectors (**6**). (**B**) Fixation parts (**9** + **10**) enable a stable stand of the bioreactor. Both parts are tightened by screws (**8**) and nuts (**11**). (**C**) Photo of the assembled bioreactor. (**D**) In silico analysis qualitatively visualizes the applied medium flow in the tissue chamber from high velocity (red) to low velocity (blue). Streamlines and arrows indicate the flow direction. (**E**) Sectional views depict the assembled bioreactor interior, i.e., the tissue chamber (T) and the medium reservoir (R).

**Figure 3 bioengineering-11-00068-f003:**
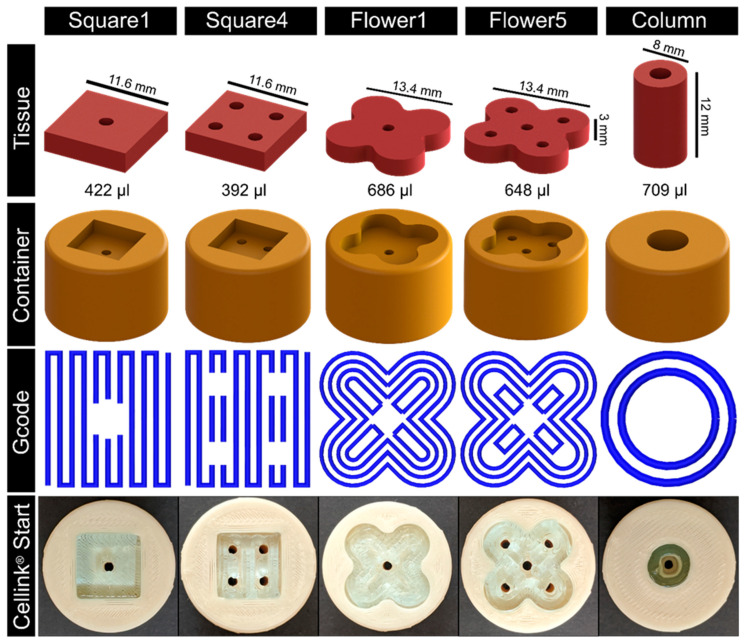
**Proof of concept by testing different geometries from tissue design to final printing assessment.** First row shows the CAD of different tissue geometries including different channel arrangements and the overall tissue volume. Second row shows the CAD of the derived tissue container. One layer of the tissue Gcode is graphically depicted in the third row. The lower two rows show the bioprinting assessment using the Cellink^®^ Start hydrogel for printing the constructs into the printed tissue container.

**Figure 4 bioengineering-11-00068-f004:**
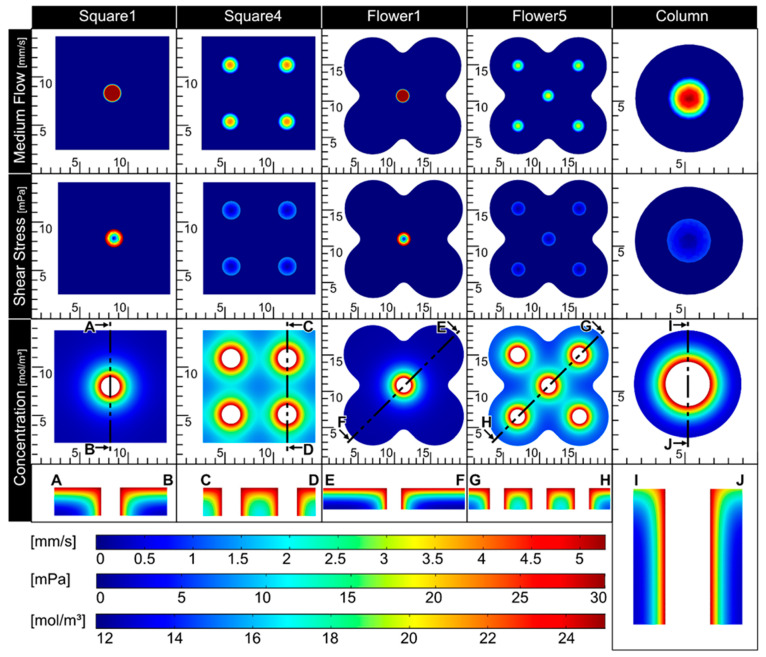
**In silico simulation of relevant parameters.** Each of the tissue geometries is assessed for the medium flow distribution of the channels (upper row), the shear stress within the channels (middle row), and the diffusion of glucose into the tissue (lower two rows). A rate of 1.5 mL/min was used as initial medium flow and the calculation of the shear stress. Initial concentration of the tissue was 0 mol/m^3^ and 25 mol/m^3^ for the medium. The diffusion coefficient was set to 6 × 10^−10^ m^2^/s [24] and the elimination rate was calculated as −1.157 × 10^−4^ mol/(m^3^·s) (glucose consumption of 3 × 10^6^ cells/mL) [25]. Glucose concentration is also shown as cut view indicated by A–B, C–D, E–F, G–H, and I–J.

**Figure 5 bioengineering-11-00068-f005:**
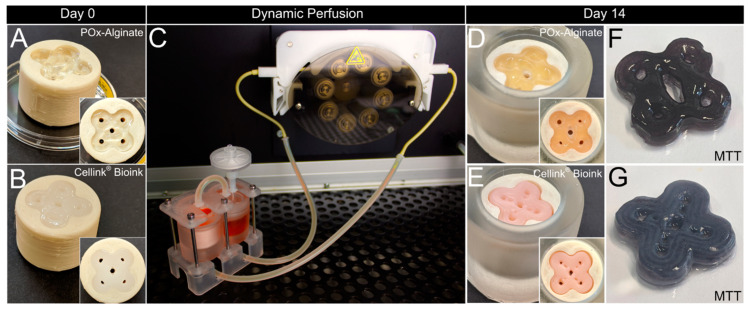
**Dynamic tissue culture and histological analysis.** Tissue construct made from POx–Alginate bioink (**A**) and Cellink^®^ Bioink (**B**) containing 1 × 10^7^ C2C12 cells per ml directly after bioprinting on day 0. (**C**) Dynamic tissue culture within bioreactor system. Tissue construct made from POx–Alginate (**D**) and Cellink^®^ Bioink (**E**) after dynamic culture for 14 days. Qualitative MTT assessment of POx–Alginate construct (**F**) and Cellink^®^ Bioink construct (**G**). Dimensions of container and tissue constructs are according to Figure 3.

**Figure 6 bioengineering-11-00068-f006:**
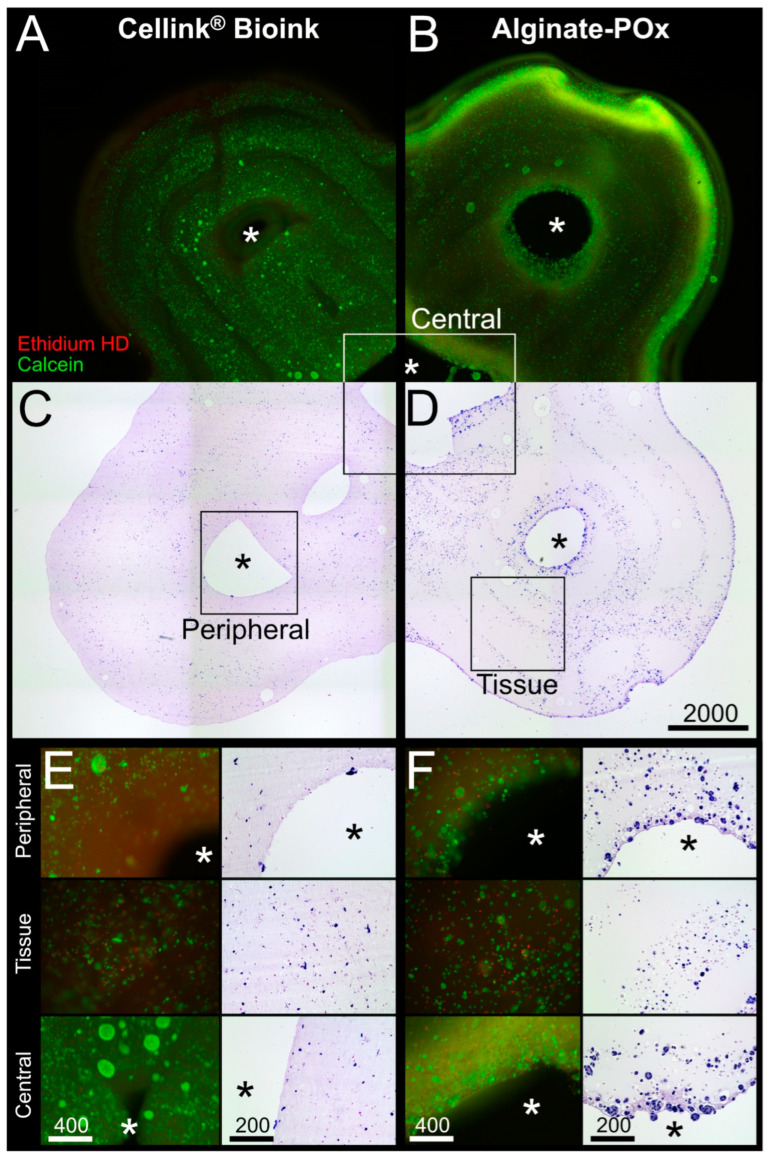
**Histological analysis.** Live–dead assay and HE staining of a tissue construct made from POx–Alginate bioink (**A**,**C**) and from Cellink^®^ Bioink (**B**,**D**), respectively, containing C2C12 cells after dynamic culture for 14 days within bioreactor system. Representative sections of areas at the peripheral pore, the tissue area, and the central pore for both bioinks (**E**,**F**). * shows the pore area. Scale bar in µm.

**Figure 7 bioengineering-11-00068-f007:**
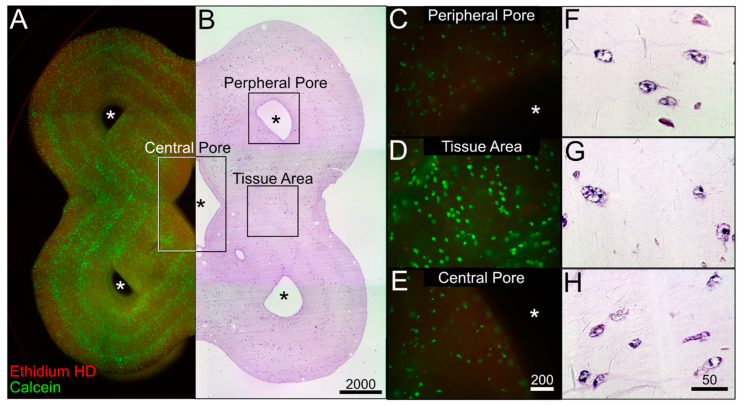
**Histological analysis.** Live–dead assay (**A**) and HE staining (**B**) of a tissue construct made from Cellink^®^ Bioink containing differentiating hMSCs after dynamic culture for 21 days within bioreactor system. Representative section of areas at the peripheral pore (**C**), the tissue area (**D**,**F**–**H**), and the central pore (**E**). * shows the pore area. Scale bar in µm.

**Figure 8 bioengineering-11-00068-f008:**
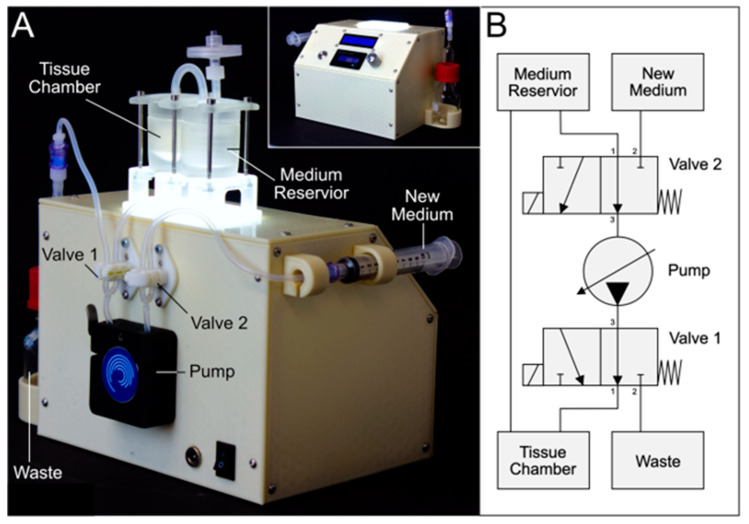
**Automated docking station for bioreactor setup.** (**A**) Isometric back view of the docking station for independent perfusion and automated media exchange (bioreactor system installed). Housing is 3D printed from lignin and electronics are based on an Arduino microcontroller. (**B**) Pneumatic circuit of the docking station and bioreactor setup.

**Figure 9 bioengineering-11-00068-f009:**
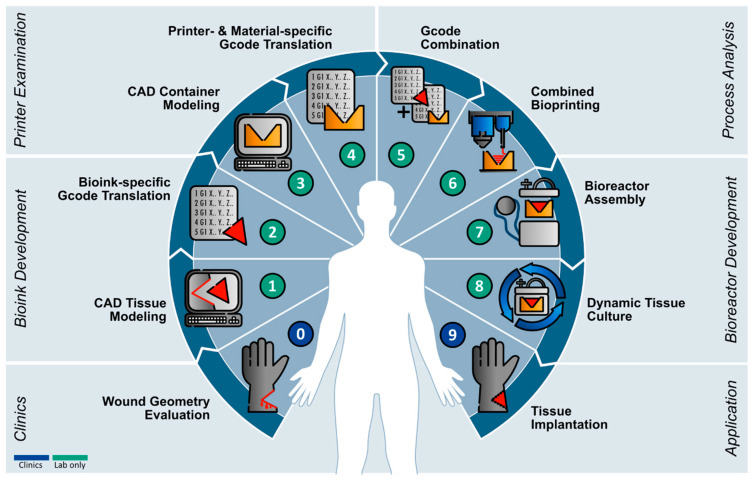
**Roadmap for defect-specific bioprinting.** Starting with geometry assessment of wound defect by, e.g., 3D scanning methods (**0**). According to wound data, the affected piece is CAD modeled (**1**) followed by translation into tissue- and printer-specific gcode (**2**). Next, the required tissue container is designed (**3**) and translated into material- and printer-specific gcodes (**4**). Gcodes are combined into a working file for individual printers (**5**). Tissue container and construct are printed by combined printing approach (**6**) and installed in the bioreactor system (**7**). Dynamic perfusion culture is applied for proper tissue maturation (**8**) until translation into the patient (**9**).

## Data Availability

Dataset available on request from the authors.

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
