# Peer review of "Perfusable Tissue Bioprinted into a 3D-Printed Tailored Bioreactor System"

_bioengineering, 2024, doi:10.3390/bioengineering11010068_

Round 1
Reviewer 1 Report
Comments and Suggestions for Authors
This paper is highly intriguing. The author successfully designed and fabricated a perfusable tissue through 3D bioprinting within a 3D-printed bioreactor. However, a major concern arises regarding the considerable size of the printed tissue, exceeding a diameter of 10mm. The critical question emerges: How can we guarantee the effective distribution of nutrients and oxygen to the central regions of the printed tissue with such a substantial size?
Minor concerns:
- 1. The scale bars on Figure 7 lack units of measurement.
- 2. It would be immensely beneficial if a video demonstration illustrating the functionality of the bioreactor could be provided in supplementary.
Author Response
Dear reviewer,
We highly appreciate your valuable input and suggestions on how to improve our paper, and hope we could answer your questions to your satisfaction. In the lower part we answered your questions point for point written in red colour.
This paper is highly intriguing. The author successfully designed and fabricated a perfusable tissue through 3D bioprinting within a 3D-printed bioreactor. However, a major concern arises regarding the considerable size of the printed tissue, exceeding a diameter of 10mm. The critical question emerges: How can we guarantee the effective distribution of nutrients and oxygen to the central regions of the printed tissue with such a substantial size?
In case a bigger tissue is needed, the system is thought to be scaled up to allow a sufficient distribution of channels for perfusion with culture medium. The nutrient supply is modeled by the preceding simulation and can be performed for any tissue shape and size. To make this clear, we have added the following in line 445 "With this, the system is scalable to bigger tissue sizes."
- The scale bars on Figure 7 lack units of measurement
"Scale bar in µm" has been added to the caption.
2. It would be immensely beneficial if a video demonstration illustrating the functionality of the bioreactor could be provided in supplementary.
We have added three videofiles to the supplementary data. S1 showing the bioprinting process into the container. S2 showing a running bioreactor with a peristaltic pump, and S3 a close up video of the media flow back into the media reservoir. With these, we hope to give a better impression of the functioning bioreactor as well as the bioprinting process.
Reviewer 2 Report
Comments and Suggestions for Authors
This manuscript describes the 3D printing of a customizable bioreactor for cell cultivation and generation of tissue models. The system was tested by bioprinting two different cell types with different geometries and the printed cells showed viabilities > 14 days. This represents a good step towards the democratization of the 3D bioprinting technology in the field of tissue engineering and biological implants. The aper is well-written and include informative figures with good quality and resolution. This reviewer recommends its publication in its current format.
Author Response
Dear reviewer,
thank you very much for your kind assessment and your recommendation for publication in its current format. We hope our edits for the other reviewers also find your approval.
Reviewer 3 Report
Comments and Suggestions for Authors
The manuscript, "Perfusable tissue bioprinted into a 3D-printed tailored bioreactor system," describes a workflow that creates a bioreactor system that could be personalized for an implant. The device is easily customizable and may support cell cultivation and tissue maturation. The topic is not novel, but it could be suitable for publication in the Bioengineering journal. However, I recommend rejecting the paper without further consideration. The main reason is that the experimental design has severe flaws, but the most critical is the need for quantitative results. Reading carefully, this is more an observational study than a research work. I recommend reconsidering to change the type of the paper or even the whole study.
Other specific comments for improving the paper in the future submissions are as follows:
1. Although the authors show cell viability images, no quantitative results are shown.
2. I suggest including more results about the differentiation process in cells. The figures should have clarified the result after 21 days.
3. Please incorporate technical information about the automated docking station for the bioreactor. Authors can use supplementary docs.
Comments on the Quality of English LanguageMinor editing of English language required
Author Response
Dear reviewer,
We highly appreciate your valuable input and suggestions on how to improve our paper, and hope that with adding a viability analysis we could adapt to your satisfaction. In the lower part we answered your suggestions point for point written in red colour.
- Although the authors show cell viability images, no quantitative results are shown.
Thank you for this suggestion. While we have shown a general survival of cells in our system, a quantification further strengthens this ability. We therefore have included live and dead cell countings in the supplementary (Supplementary Figure S2). For this, representative areas from the whole tissue were counted. Cell viability is given for all cell types and bioinks, showing good viability in each approach.
2. I suggest including more results about the differentiation process in cells. The figures should have clarified the result after 21 days
While figure 6 refers to C2C12 cells, we wanted to show here that cultivation of cells is generally possible. In a next step, we wanted to show the ability of our system to differentiate MSC into adipogenic lineage in general and cell survival over 21 days. For this proof of concept, visible cytoplasmic vacuoles with lipids represented a sufficient result for the successful differentiation of early adipocytes. Further tests, suchs as oil red staining might also have been useful.
3. Please incorporate technical information about the automated docking station for the bioreactor. Authors can use supplementary docs.
We have included a circuit diagram and a flow diagram of the docking station in the supplementary files (Supplementary Figure S3).
Reviewer 4 Report
Comments and Suggestions for Authors
This is an interesting study. Overall the conclusion is supported by the data. I only have some specific comments:
1, what's the reason of using Alginate-POx bioink?
2, why the cell culture density of C2C12 and MSC are different?
3, It will be great if scale bar is added in figure 5.
4, Please provide quantitative results for the live-dead assay conducted in figure 6 and 7.
Author Response
Dear reviewer,
We highly appreciate your valuable input and suggestions on how to improve our paper, and hope we could answer your questions to your satisfaction. In the lower part we answered your questions point for point written in red colour.
1,what's the reason of using Alginate-POx bioink?
We thank the reviewer for this comment and chance for us to improve clarity. We chose to include the Alginate-POx bioink, as we showed previously that excellent 3D printability of this hybrid bioink. For us, it was particularly interesting to compare our experimental bioink with a commercial one. (C. Hu, Biofabrication 2022, 14, 025005. DOI: 10.1088/1758-5090/ac40ee.). While commercial inks are easier available, they are more expensive, and less versatile and adaptable (e.g. pore sizes). In our case, Alginate-POx did show a better printing behaviour. To make the versatility point more clear, we have added the following in lines 462ff "The POx-Alginate was included in this study to allow a head-to-head comparison with the commercial bioink. Previously, we have shown that the combination of alginate and the POx based diblock copolymers hydrogels significantly improves 3D printability of alginate and allows for long-term cell culture [23]. Now, we wanted to compare directly with a commercial bioink, to gauge the potential of our experimental bioink." And in lines 470f "A blended bioink has the advantage to be easier adaptable to a specific use case (e.g. temperature, viscosity, or pore size)"
2, why the cell culture density of C2C12 and MSC are different?
This was decided due to the availability of the cells. While C2C12 is an immortalized cell line, numbers are of no concern. MSCs, however, are primary cells that have limited use time. Generally, a minimum of 3*10^6 MSC/ml has been established in our institute for the generation of adipose tissue models, which we even exceeded. Tissues of the size that we used need high cell numbers, which makes it costly.
3, It will be great if scale bar is added in figure 5.
Here we decided to not include a scale bar as container and tissue size is already given in Figure 3. To make this clearer, we have added the following to the description of figure 5 "Dimensions of container and tissue constructs are according to Figure 3."
4, Please provide quantitative results for the live-dead assay conducted in figure 6 and 7.
Thank you for this suggestion. While we have shown a general survival of cells in our system, a quantification further strengthens this ability. We therefore have included live and dead cell countings in the supplementary. For this, representative areas from the whole tissue were counted. Cell viability is given for all cell types and bioinks, showing good viability in each approach. The figure can be found in the Supplementary Figure S2.
Round 2
Reviewer 3 Report
Comments and Suggestions for Authors
Accept in present form as authors submitted some quantitative results in suppl. material.